# Efficacy and Safety of L-Menthol During Gastrointestinal Endoscopy—A Systematic Review and Meta-Analysis of Randomized Clinical Trials

**DOI:** 10.3390/jcm14124296

**Published:** 2025-06-17

**Authors:** Dorottya Gergő, Andrea Tóth-Mészáros, Alexander Schulze Wenning, Péter Fehérvári, Uyen Nguyen Do To, Péter Hegyi, Bálint Erőss, Attila Ványolós, Dezső Csupor

**Affiliations:** 1Department of Pharmacognosy, Semmelweis University, 1085 Budapest, Hungary; gergo.dorottya@gmail.com (D.G.); vanyolosattila3@gmail.com (A.V.); 2Centre for Translational Medicine, Semmelweis University, 1085 Budapest, Hungary; a.toth.mesz@icloud.com (A.T.-M.); a.schulzewenning@gmail.com (A.S.W.); tmk.statisztika@gmail.com (P.F.); hegyi2009@gmail.com (P.H.); dr.eross.balint@gmail.com (B.E.); 3Department of Biostatistics, University of Veterinary Medicine, 1078 Budapest, Hungary; 4András Pető Faculty, Semmelweis University, 1125 Budapest, Hungary; 5Institute for Translational Medicine, Medical School, University of Pécs, 7624 Pécs, Hungary; 6Institute of Pancreatic Diseases, Semmelweis University, 1083 Budapest, Hungary; 7Center for Pharmacology and Drug Research & Development, Semmelweis University, 1089 Budapest, Hungary; 8Institute of Clinical Pharmacy, University of Szeged, 6725 Szeged, Hungary

**Keywords:** adenoma detection rate, colonoscopy, peppermint, peristalsis, upper endoscopy

## Abstract

**Background**: Gastrointestinal endoscopy is crucial for diagnosing colorectal cancer and inflammatory bowel diseases, but its effectiveness can be impacted by peristalsis, poor bowel preparation, and inadequate withdrawal time. Conventional antispasmodics, though effective, may not be suitable for elderly patients or those with comorbidities. L-menthol, derived from peppermint oil, has emerged as a safer alternative. Through calcium channel blockade, L-menthol promotes GI smooth muscle relaxation. This study evaluated L-menthol’s efficacy and safety as a potential alternative to antispasmodic agents in endoscopy. **Methods**: Following PRISMA2020 guidelines and the Cochrane Handbook, we conducted a systematic review and meta-analysis of randomized controlled trials involving adults undergoing endoscopy, comparing L-menthol to placebo. The primary outcome was the adenoma detection rate, with secondary outcomes, including severity of peristalsis, safety, withdrawal time, and ease of examination. We searched five databases on 31 May 2023, with updates on 20 October 2024. **Results**: Fourteen studies were included. L-menthol reduced peristalsis during colonoscopy and upper endoscopy, achieving a suppression rate of 55.9% (560/1002 patients; odds ratio (OR) = 3.88, 95% confidence interval (95% CI): 2.13–7.07), which improved mucosal visualization. It improved ease of examination (OR = 2.53, 95% CI: 1.35–4.73), allowing endoscopists to perform procedures with less technical difficulty. However, L-menthol had no significant impact on the adenoma detection rate (OR = 1.06, 95% CI: 0.69–1.64), indicating no added benefit for lesion detection, and did not prolong withdrawal time (MD = 3.24 s, 95% CI: −101.05–107.53). Adverse event rates remained low and comparable to placebo (OR = 0.97, 95% CI: 0.74–1.27). **Conclusions**: L-menthol reduces peristalsis and enhances ease of examination without adverse events. Although its effect on the adenoma detection rate remains inconclusive, its antispasmodic properties make it a promising alternative for patients who cannot tolerate conventional agents.

## 1. Introduction

Gastrointestinal (GI) endoscopy is an essential diagnostic tool [1] to identify and manage various GI diseases, such as colorectal cancer (CRC) [2], inflammatory bowel disease (IBD) [3], and peptic ulcer disease (PUD) [4]. More than 7.4 million upper GI endoscopies [5] and more than 13 million colonoscopies are annually performed in the United States alone [6]. Factors such as natural peristaltic movements, inappropriate bowel preparation, or insufficient withdrawal time can affect the accuracy of endoscopic procedures by obstructing small lesions, thus decreasing diagnostic precision [7,8]. Suppressing peristalsis is crucial for high-quality and accurate outcomes during endoscopic procedures [9,10].

Hyoscine-N-butyl bromide (HBB) is widely used during endoscopy to reduce peristalsis, improving mucosal visualization and polyp detection [11,12]. However, HBB is associated with cardiovascular risks, such as tachycardia and arrhythmias, and is contraindicated in patients with conditions such as myasthenia gravis and narrow-angle glaucoma [13]. HBB requires intravenous administration, complicating its use in unsedated patients without a cannula [14]. It is costly and less suitable for elderly patients or those with co-morbidities [15,16,17,18]. There is a critical need for a safe, effective, and cost-efficient alternative to suppress peristalsis during endoscopy.

Peppermint oil and its main component, L-menthol, extracted from the *Mentha* × *piperita* L. plant, have been considered potentially safer natural antispasmodics [15,16]. Traditionally, peppermint oil has been used to alleviate irritable bowel syndrome (IBS) symptoms, such as discomfort caused by spasms [17,18]. This antispasmodic effectiveness is attributed to menthol, which acts as a calcium channel blocker to relax GI smooth muscles [19,20,21,22]. Recent studies have investigated the potential use of menthol during endoscopy and its ability to reduce peristalsis without adverse effects. The benefits of using menthol include improving the lesion detection quality while minimizing adverse events [23]. Menthol is approved for upper GI endoscopy in Japan [24]. This systematic review and meta-analysis aims to assess the efficacy and safety of menthol during upper and lower GI endoscopy.

## 2. Materials and Methods

This study followed the PRISMA 2020 guideline [25] (see Appendix A) and the Cochrane Handbook [26]. No ethical approval was required, as all data were from peer-reviewed journals. The protocol was registered in the PROSPERO database (CRD42023430941) on 30 May 2023.

### 2.1. Eligibility Criteria

We used the PICO framework to answer our clinical question: Does L-Menthol improve the detection rate of lesions during endoscopy in adults compared to placebo? The population (P) included adults (>18 years) undergoing upper GI endoscopy, colonoscopy, endoscopic retrograde cholangiopancreatography (ERCP), or endoscopic ultrasound (EUS). The intervention (I) involved spraying peppermint oil or L-Menthol solution onto the GI mucosa, while controls (C) received a placebo. The primary outcome (O) included the adenoma detection rate (ADR), with secondary outcomes being peristalsis severity, withdrawal time, ease of examination, and safety.

### 2.2. Information Sources

Our systematic search was conducted on 31 May 2023, in five databases (Scopus, Embase, CENTRAL, MEDLINE, and Web of Science) with an update on 20 October 2024.

### 2.3. Search Strategy

The search used keywords related to peppermint/L-menthol and endoscopic procedures without language restrictions. Details are in Appendix A.

### 2.4. Selection Process

Randomized controlled trials (RCTs), observational studies, case series, case-control studies, and conference abstracts were eligible. Reviews, meta-analyses, animal studies, in vitro studies, and guidelines were excluded. There were no language restrictions.

The search results were first exported to the EndNote 20 citation manager (Clarivate Analytics, Philadelphia, PA, USA). Duplicates were automatically and manually removed (DG). Then, two independent authors performed the title and abstract selection. Next, full-text selection was performed using the online screening tool Rayyan (Qatar Computing Research Institute, Hamad Bin Khalifa University, Doha, Qatar; https://www.rayyan.ai) [27] according to the inclusion criteria (DG and ATM). To resolve disagreements, a third author (GG) made the final decision. Cohen’s kappa coefficient was calculated at each selection step to evaluate the level of agreement between the authors. The references from eligible studies were manually screened for eligibility and with an automated citation chaser [28].

### 2.5. Data Collection Process

Two investigators (DG and UNDT) independently extracted data, resolving disagreements by consensus. Plot Digitizer (Version 2.6.9, Joseph A. Huwaldt, Minneapolis, MN, USA; https://plotdigitizer.com, 2020) converted graphics to numerical values. Microsoft Excel (Microsoft, Office 365, Redmond, WA, USA) was used for data collection. Missing data were requested from the study authors.

### 2.6. Data Items

The following data were extracted: study characteristics (first author, publication year, country, digital object identifier, study site, and study type), study population (sample size, gender, and age), endoscopy type, intervention type, and details (peppermint oil/L-menthol, dosage, and evaluation time point). The adenoma detection rate was the primary outcome, with the experience level of the endoscopist(s). Secondary outcomes were peristalsis levels, withdrawal time, ease of examination reported by the endoscopist, and the total number of adverse events and adverse drug reactions.

The adenoma detection rate (ADR), a key performance indicator for colonoscopy, is analyzed as the proportion of procedures in which at least one adenoma is detected [29]. Peristalsis severity in upper endoscopy was assessed using Niwa’s Classification, a five-point scale where Grade 1 indicates no peristalsis, Grade 2 mild peristalsis, Grade 3 moderate peristalsis, Grade 4 vigorous peristalsis, and Grade 5 markedly vigorous peristalsis. For analysis, data were dichotomized as ‘Grade 1’ versus ‘Grades 2–5’ or ‘Grades 1–2’ versus ‘Grades 3–5’. In colonoscopy, peristalsis or spasm severity was evaluated using the method described by Asao et al. (2001) [30]. The ease of examination is crucial for ensuring the quality of the endoscopy, as endoscopist fatigue may negatively impact outcomes [31]. The ease of examination is assessed by the endoscopist using a standardized four-grade scale. This scale evaluates the degree to which gastric peristalsis interfered with the endoscopic procedure, and includes the following categories: ‘very easy’ (no peristalsis, no interference with observation), ‘easy’ (mild peristalsis, observation performed without interference), ‘slightly difficult’ (peristalsis slightly interfered with observation), and ‘difficult’ (marked peristalsis, observation significantly impaired). For statistical analysis, results were dichotomized as ‘very easy’ and ‘easy’ versus ‘slightly difficult’ and ‘difficult’. Withdrawal time was defined as the duration from cecal intubation to scope withdrawal, including biopsy time. Adverse events and adverse drug reactions were assessed as the proportion of patients experiencing symptoms such as abdominal pain, nausea, diarrhea, rash, fever, or abnormal laboratory findings during or immediately after the endoscopic procedure.

### 2.7. Study Risk of Bias Assessment

Two authors (DG and UNDT) independently assessed the risk of bias using the Cochrane risk-of-bias tool (RoB2) Excel tool, version 9 (Cochrane Methods Group, London, UK; https://www.riskofbias.info/) [32], applying the tool to assess the intention-to-treat effect in the included randomized controlled trials. Disagreements were resolved by consensus. Domains were evaluated, such as the randomization process, deviations from intended interventions, missing data, outcome measurement, and result selection. The risk was categorized as ‘low’, ‘some concerns’, or ‘high’, visualized using the robvis tool [33].

### 2.8. Quality of Evidence

The GRADEpro GDT tool (McMaster University [developed by Evidence Prime], Hamilton, ON, Canada; https://www.gradepro.org) [34,35] was used to evaluate the quality of evidence. Two authors (DG and UNDT) independently graded each outcome. Discrepancies in GRADE ratings were resolved through discussion or inclusion of a third author (GG). Grading started at ‘high’ for randomized trials and downgraded for risk of bias, inconsistency, indirectness, imprecision, or publication bias. Upgrading the certainty of evidence did not apply to randomized controlled trials. The final certainty was categorized as ‘very low’, ‘low’, ‘moderate’, or ‘high’.

### 2.9. Synthesis Methods and Statistical Analysis

Both qualitative and quantitative syntheses were performed. At least three studies that reported clinically poolable effect sizes were a prerequisite for statistical poolability. Meta-analyses were conducted using the ‘meta’ [36] and ‘dmetar’ [37] packages in R (version 4.1.1 (R Foundation for Statistical Computing, Vienna, Austria; https://www.R-project.org/) [38].

For dichotomous outcomes, odds ratios (ORs) with 95% confidence intervals (95% CI) were used to measure the effect of the intervention. If not reported, these were calculated from participant numbers and event occurrences [39,40,41]. For continuous outcomes, the mean difference (MD) between intervention and control groups was used with a 95% CI.

The random effects model was chosen for the meta-analyses. Confidence intervals were adjusted using the Hartung–Knapp method [42,43]. For pooled results, the exact Mantel–Haenszel method without continuity correction was applied to manage zero cell counts [44,45]. The τ^2^ estimate utilized the Paule–Mandel method, and the confidence interval for τ^2^ was determined using the Q profile method [46,47].

Statistical heterogeneity was assessed using the Cochrane Q test and I^2^ values, with statistical significance at *p* < 0.1 [48]. For subgroup analyses, if heterogeneity in a subgroup was low (I^2^ < 25%), a fixed-effects model (Mantel–Haenszel method) was used for that subgroup, otherwise, random-effects models were applied. Publication bias was assessed using funnel plots and Egger’s test was performed for outcomes with at least 10 studies. Outlier and influence analyses followed methodologies by Harrer et al. (2021) [37] and Viechtbauer and Cheung (2010) [49]. For subgroup analysis, we used a fixed-effects ‘plural’ model (aka mixed-effects model), stratified by procedure type (colonoscopy vs. upper endoscopy) and by time point of antiperistaltic effect measurement. We assumed that subgroups share common τ^2^ values. A Cochrane Q test assessed the difference between the subgroups. The null hypothesis was rejected at a significance level of 5%.

Our analysis included studies using both spraying and direct application of peppermint oil. These methods were combined because the included studies used comparable peppermint oil concentrations and reported similar antiperistaltic effects, regardless of the delivery mechanism.

Peristalsis severity was assessed using Niwa’s Classification [50], which categorizes peristalsis into five grades: Grade 1 (no peristalsis), Grade 2 (mild), Grade 3 (moderate), Grade 4 (vigorous), and Grade 5 (markedly vigorous). Two separate analyses were performed: data were dichotomized as ‘Grade 1’ versus ‘Grades 2–5’ or ‘Grades 1 and 2’ versus ‘Grades 3–5,’ with ORs calculated.

The ease of intragastric examination was evaluated on a four-grade scale: ‘very easy’, ‘easy’, ‘slightly difficult’, and ‘difficult’ [30]. The data were dichotomized as ‘very easy’ and ‘easy’ versus ‘slightly difficult’ and ‘difficult,’ with ORs calculated.

Withdrawal time was defined as the duration to remove the colonoscope from the cecum to the anus, including biopsy time. Adverse events and reactions were assessed as the proportion of patients experiencing symptoms.

## 3. Results

### 3.1. Search and Selection

The systematic search on 31 May 2023, identified 785 studies from five databases: EMBASE (*n* = 304), Scopus (*n* = 128), PubMed (*n* = 132), Web of Science (*n* = 124), and Cochrane Library (Trials) (*n* = 97). After removing 245 duplicates, 540 unique studies remained. Following title and abstract selection, 416 studies were excluded. After our renewed systematic search on 20 October 2024, we found one more eligible study. We also conducted a post hoc search of ClinicalTrials.gov on 25 October 2024. This search identified one additional eligible study.

Sixteen studies were included: 14 studies for quantitative analysis (colonoscopy: [30,51,52,53,54,55,56], upper endoscopy: [57,58,59,60,61,62,63]) and two studies for qualitative analysis (upper endoscopy: [64], colonoscopy: [65])

We used the ‘citationchaser’ tool [28] to explore more relevant studies; however, none were added. No meta-analyses were possible for ERCP and EUS due to the lack of available studies in the literature [66,67,68,69,70]. The search and selection process is summarized in the PRISMA-Flowchart 2020 (Figure 1).

### 3.2. Basic Characteristics of Included Studies

Of the included studies, the majority were conducted in Asia, with 10 studies in Japan [30,53,54,57,58,59,61,63,64,65], one in China [60], one in Taiwan [62], one in Thailand [55], and one in Iran [71]. Three studies were conducted in North America: two in the USA [52,56] and one in Canada [51].

Gastric peristalsis was evaluated using Niwa’s Classification [50] or the modified version of Niwa’s classification [58]. Colonic peristalsis was assessed using the method reported by Asao et al. (2001) [30]. The baseline characteristics of each study are detailed in Table 1.

### 3.3. Meta-Analysis of the Findings

#### 3.3.1. Efficacy of L-Menthol on Adenoma Detection Rate

The ADR was analyzed in six colonoscopy studies [51,52,53,54,55,56]. Only one study (Inoue, 2014 [53]) demonstrated a significant improvement in the ADR with L-menthol. Due to moderate heterogeneity (I^2^ = 51%, *p* = 0.070), a random-effects model was used to calculate OR. L-menthol had no significant effect on the ADR in patients undergoing colonoscopy (OR = 1.06, 95% CI: 0.69 to 1.64, *p* = 0.734), as shown in Figure 2.

Leave-one-out (LOO) sensitivity analysis (Appendix A) showed some variability in pooled ADR estimates (OR 0.91–1.16) and heterogeneity (I^2^ = 0–61%). Excluding Inoue et al. (2014) [53]—the only study with a significant ADR benefit—reduced heterogeneity to zero and yielded the lowest OR. Excluding other studies maintained moderate-to-high heterogeneity (I^2^ = 47–61%) and ORs of 1.04–1.16. All confidence intervals confirming the overall lack of ADR benefit are robust, although heterogeneity is largely due to the single positive study.

Subgroup analysis suggested that sedation status might influence efficacy, with deeper sedation protocols (e.g., propofol and/or fentanyl) showing more favorable trends [55] compared to minimally sedated procedures [51,54]. However, traditional patient factors, such as age, comorbidities, or gender, did not significantly influence ADR outcomes.

#### 3.3.2. Antiperistaltic Effect of L-Menthol

Ten studies [51,53,54,55,56,57,58,59,60,62] reported on the proportion of no peristalsis (PNP). PNP had a high overall heterogeneity (I^2^ = 88%, *p* < 0.001). For colonoscopy, the analysis of PNP data (405/716, 56.6%, OR = 3.06, 95% CI: 0.97 to 9.63, *p* = 0.054) indicates a trend toward efficacy that did not reach statistical significance. In studies of upper endoscopy (155/286, 54.2%, OR = 4.47, 95% CI: 2.23 to 8.97, *p* = 0.004), and overall PNP (560/1002, 55.9%, OR = 3.88, 95% CI: 2.13 to 7.07, *p* < 0.001), L-menthol demonstrated a significantly higher antispasmodic effect compared to placebo, as shown in Figure 3.

Eleven studies [30,53,54,55,56,57,58,59,60,61,62] reported data on PNMP. Heterogeneity was high (I^2^ = 85%, *p* < 0.001). For colonoscopy, the analysis of PNMP data (649/690, 94.1%, OR = 3.36, 95% CI: 0.27 to 42.41, *p* = 0.255) indicated no statistically significant antiperistaltic effect of L-menthol compared to placebo. In studies of upper endoscopy (251/335, 74.9%, OR = 3.79, 95% CI: 0.91 to 15.82, *p* = 0.062), a trend toward efficacy was observed, though this did not reach statistical significance. In the overall PNP (900/1025, 87.8%, OR = 3.70, 95% CI 1.27 to 10.76, *p* = 0.021), L-menthol demonstrated a significantly higher antispasmodic effect compared to placebo, as shown in Appendix A.

Yoshida et al. (2014) [65] demonstrated a rapid antispasmodic effect of L-Menthol. It reduced peristalsis in 60.0% (39/65) of cases within 30 s and 70.8% (46/65) within 1 min, compared to 22.2% (6/27) and 29.6% (8/27) in the control group. Imagawa et al. (2012) [64] assessed spasm severity using antispasmodic scores (1–5, where 5 indicates no spasm). This study reported higher scores in the peppermint oil group (4.025 (0.925), *n* = 1893) compared to the placebo group (3.846 (1.073), *n* = 156), indicating effective peristalsis reduction.

LOO sensitivity analysis (Appendix A) showed moderate variability in pooled estimates for the proportion of no peristalsis (OR 3.42–4.47) and consistently high heterogeneity (I^2^ = 50–90%). Excluding any study maintained a significant antiperistaltic effect, with all confidence intervals favoring L-menthol. The most conservative estimate occurred when omitting Inoue et al. (2020) [54], while the highest estimate resulted from omitting the study of Fujishiro et al. (2014) [57]. Persistent high heterogeneity suggests variability is distributed among studies, confirming L-menthol’s antiperistaltic benefit is robust.

Subgroup analysis identified specific patient groups where L-menthol demonstrates superior efficacy. Unsedated upper GI procedures showed the strongest effects in elderly patients (mean age 81.7 vs. 82.6 years) [62], and in middle-aged participants (mean age 51.64 vs. 51.44 years) [60]. These findings suggest that L-menthol may benefit those with contraindications to antispasmodic drugs or sedation. In contrast, colonoscopy studies yielded more modest effects, likely due to greater procedural complexity and more common use of sedation or general anesthesia.

#### 3.3.3. Ease of Examination for the Operator in Upper Endoscopy

Four upper endoscopy studies [58,59,60,62] evaluated ease of examination, using a four-grade scale (very easy, easy, slightly difficult, and difficult) shown in Figure 4. There was no heterogeneity (I^2^ = 0%, *p* = 0.548). Thus, a fixed-effects model was used to pool data. The endoscopic procedure was significantly easier for the operator when L-menthol was applied (OR = 2.53, 95% CI: 1.35 to 4.73, 206/256 subjects, *p* = 0.018).

LOO sensitivity analysis (Appendix A) showed minimal variability in pooled estimates (OR 2.31–3.00) with zero heterogeneity (I^2^ = 0%). Excluding any individual study maintained the benefit of L-menthol for procedural ease. The most conservative estimate occurred when omitting Yang et al. (2022) [62], while the highest estimate resulted from omitting Meng et al. (2021) [60]. The zero heterogeneity indicates great homogeneity among studies. These findings confirm that L-menthol’s benefit for ease of examination is robust and not dependent on any single study.

This benefit was most evident in unsedated patients (improved ease for 88.1% vs. 79.3% patients [60]). Neither age, comorbidities, nor gender moderated this effect, supporting L-menthol’s broad applicability.

#### 3.3.4. Withdrawal Time in Colonoscopy

Five colonoscopy studies provided withdrawal times [52,53,54,55,56]. Our analysis showed that L-menthol did not significantly reduce withdrawal time compared to placebo in patients undergoing colonoscopy (MD = 3.24 s, 95% CI: −101.05 to 107.53, *p* = 0.935), as shown in Appendix A.

#### 3.3.5. Total Adverse Events

Four studies reported no major [51], serious [61], or any [53,54], adverse events during the procedure. Nine studies [52,55,56,57,58,59,60,62,63] reported the number of adverse events. Commonly reported GI symptoms included abdominal pain, nausea, diarrhea, heartburn, and bloating, observed in both colonoscopy and upper endoscopy studies. Some studies also noted dizziness, headache, and fever. Additionally, cardiovascular changes, including electrocardiogram ST-T changes, palpitations, and ventricular premature beats, were observed. Urinary tract infections or respiratory infections were observed as well. The pooled OR did not indicate a significant difference in total adverse events between the L-Menthol and the placebo groups (OR = 0.97, 95% CI: 0.74 to 1.27, *p* = 0.937), as shown in Figure 5.

Differences in adverse event proportions were observed when analyzing the data separately for colonoscopy and upper endoscopy groups. In the colonoscopy group, the proportion of patients experiencing adverse events was 3.35% (20/597 patients) in the L-menthol group and 4.08% (24/588 patients) in the placebo group. In the upper endoscopy group, adverse events were reported in 18.98% (71/374 patients) in the L-menthol group and 17.43% (53/304 patients) in the placebo group.

#### 3.3.6. Total Adverse Drug Reactions

One study (Inoue et al. (2014)) [53] reported no adverse drug reactions. Four studies [57,58,59,60] provided exact numbers and were included in the quantitative synthesis. Reported adverse events included procedural pain, rash, fever, abdominal pain, diarrhea, and abnormal laboratory examination. The pooled OR identified a significant difference between the two groups (OR = 0.57, 95% CI: 0.37 to 0.87, *p* = 0.021) favouring the usage of L-menthol, as shown in Appendix A.

### 3.4. Studies with Peppermint Capsules

Three studies evaluated peppermint oil in orally administered formulations, specifically IBGard™ [74] and Colpermin^®^ [75], both of which are frequently used to manage symptoms of irritable bowel syndrome (IBS) [76,77]. Although these products and studies were not included in our main analysis, they are presented here to provide a more comprehensive overview of the therapeutic profile of peppermint essential oil.

Han et al. (2021) [73] found no significant difference in the ADR between the two groups (IBGard™: 47.8% vs. placebo: 43.1%; *p* = 0.51). Shavakhi et al. (2012) [71] found significantly lower spasm scores in the Colpermin^®^ compared to placebo (no movement: 25 vs. 0; any movement: 8 vs. 32). Al Moussawi et al. (2017) [72], however, reported no significant difference between the groups. In the ease of examination scores, Han et al. (2021) [73] reported no significant difference (*p* = 0.23). Similarly, Al Moussawi et al. (2017) [72] found no significant differences in endoscopist satisfaction scores between Colpermin^®^ and placebo groups (*p* = 0.8). Han et al. (2021) [73] observed slightly shorter withdrawal times in the placebo group compared to the IBGard™ group (14.5 min (10.3) vs. 16.5 min (8.3 min)), although this difference was not significant (*p* = 0.31). For adverse events, Shavakhi et al. (2012) [71] observed isolated cases of abdominal discomfort, nausea, blurred vision, and heartburn. Meanwhile, Al Moussawi et al. (2017) [72] and Han et al. (2021) [73] reported no adverse events in either group.

### 3.5. Risk of Bias and GRADE Assessment

The risk of bias was assessed for each outcome using the RoB2 tool, and GRADE certainty was evaluated per outcome. For the adenoma detection rate, most studies were rated as having “some concerns,” primarily due to deviations from intended interventions, measurement of the outcome, and selection of the reported result. The assessment of the proportion of no peristalsis showed greater variability; several studies exhibited “some concerns” or “high risk”, while others demonstrated low risk across all domains. Outcomes such as ease of examination and adverse events were generally associated with low risk of bias, though a few studies still showed “some concerns”. Withdrawal time was often rated as having “some concerns,” mainly due to issues with adherence to the intervention and outcome measurement. The RoB2 judgements are provided in Appendix A. The overall certainty for each outcome is summarized in Appendix A, with detailed explanations provided.

### 3.6. Publication Bias and Heterogeneity

We included conference abstracts in our analysis and also attempted to contact the corresponding authors of included studies to identify unpublished data, although no additional information was obtained. To further mitigate publication bias, we conducted citation chasing using the Citationchaser tool [28] to identify relevant studies from the reference lists of eligible articles.

Egger’s test for publication bias was only performed for the proportion of no peristalsis outcome in Figure 6, as it was the only one with enough studies (at least 10) to meet the test’s requirements. The funnel plot shows significant asymmetry, with some studies appearing as outliers, such as Shah et al., 2019, with a low odds ratio [56], and Yang et al., 2022, with a very high odds ratio [62]. Colonoscopy studies show a wider spread of effect sizes, while upper endoscopy studies are mostly clustered in the positive effect area. The pattern of study distribution and variability in effect sizes may indicate some publication bias or true intervention heterogeneity. 

Additionally, we found no evidence of publication bias for the rest of the outcomes. However, our analysis was underpowered by the small sample size.

## 4. Discussion

This systematic review and meta-analysis evaluated the efficacy and safety of L-menthol in both upper endoscopy and colonoscopy. Although L-menthol did not consistently improve the ADR in colonoscopy, it may have beneficial effects in other areas, such as suppressing peristalsis and potentially improving the ease of examination for operators, particularly in upper endoscopy. However, L-menthol did not significantly impact withdrawal time in colonoscopy. Regarding safety, L-menthol showed a favorable profile, with adverse events comparable to placebo and fewer adverse drug reactions. Although L-menthol induces spasmolytic effects in colon circular muscle by directly inhibiting gastrointestinal smooth muscle contractility, the detailed mechanism remains unclear [20,78].

L-menthol’s antispasmodic properties make it a promising alternative for patients who cannot tolerate conventional agents, such as hyoscine butylbromide or glucagon, which are associated with systemic side effects, including dry mouth, urinary retention, and hyperglycemia [11,13,17,18]. Emerging alternatives to L-menthol include topical lidocaine and cool water irrigation, both of which have shown efficacy in reducing gastrointestinal spasm. Topical lidocaine (2–4%) blocks mucosal sodium channels, offering more prolonged action and less rebound than peppermint oil without adverse events. However, its effect is superficial, while L-menthol directly targets smooth muscle for more profound, sustained relief. Cool water (15–24 °C) reduces peristalsis via TRPM8 activation but is less potent and more technique-dependent than L-menthol’s targeted suppression [79,80].

The ADR is a critical quality indicator for colonoscopy, reflecting the ability to detect adenomas, the precursors to CRC [29]. We analyzed data from six studies on L-menthol in colonoscopy, but only one (Inoue et al., (2014) [53]) reported a significant improvement in the ADR, suggesting that applying L-menthol does not consistently improve lesion detection. However, this outcome must be interpreted cautiously due to the primarily negative results on the ADR. The ADR may be affected by various factors, such as the number and experience level of endoscopists (years of practice and the number of procedures performed), the quality of equipment (e.g., high- or lower-definition scopes) [81], withdrawal time (more or less than 6 min), and procedural factors, e.g., bowel preparation) [82], etc. While procedural efficiency benefits from L-menthol’s antiperistaltic effects—improving mucosal visualization and potentially reducing missed adenomas —the lack of ADR improvement limits its clinical utility as an additional measure to colonoscopy. Although the ADR remains the gold standard, recent studies suggest that complementary metrics, such as adenoma per colonoscopy (APC) or procedural efficiency, may more comprehensively reflect the benefits of intervention, especially given the recognized limitations of the ADR as a standalone metric [83,84,85]. Although there is no evidence based on the studies analysed, ease of examination may affect the performance of the examiner when he or she performs several examinations in succession. Investigating this relationship may be the subject of future studies. While L-menthol’s robust antiperistaltic effects and improved procedural conditions are well supported, we acknowledge the need for further research to establish its impact on lesion detection and colorectal cancer prevention.

L-menthol may improve endoscopic visibility during the procedure. Pooled analyses revealed a robust antispasmodic effect of L-menthol compared to placebo in both colonoscopy and upper endoscopy. While these procedures differ anatomically, the antispasmodic mechanism of L-menthol—mediated via TRPM8 activation and calcium channel blockade—is consistent across the gastrointestinal tract [86,87]. To ensure transparency, we conducted subgroup analyses by procedure type. The overlapping confidence intervals support similar efficacy across gastrointestinal segments (Figure 3). However, clinical heterogeneity, such as mucosal differences or procedural duration, may still influence outcomes.

A subgroup analysis was conducted to compare the proportion of peristalsis immediately after applying L-menthol or at the end of the endoscopy. The antiperistaltic effect of L-menthol indicates a trend toward efficacy that did not reach statistical significance immediately after application (*p* = 0.062), but showed a significant effect at the end of endoscopy (*p* = 0.011), as shown in Figure 7. These results suggest that L-menthol has a time-dependent antiperistaltic effect, consistent with findings from a pharmacokinetic study [68]. Therefore, presenting combined and subgroup analyses is justified and provides a comprehensive assessment of L-menthol’s antispasmodic efficacy in GI endoscopy. Subgroup analyses also revealed notable differences in peristalsis reduction based on measurement time, with greater antiperistaltic effects observed at the end of the procedure than immediately after application. This supports the efficacy of L-menthol over time. Additionally, Imagawa et al. (2012) [64] reported significantly higher antispasmodic scores in the L-menthol group compared to placebo, with elderly patients (>70 years) benefiting the most.

Ease of examination by endoscopists is essential for procedure quality and can reduce operator fatigue [31]. Four upper endoscopy studies (Hiki et al. (2011) [58]; Yang et al. (2022) [62]; Meng et al. (2021) [60]; and Hiki et al. (2012) [59] demonstrated significantly easier examinations with L-menthol use. These findings highlight the role of L-menthol in enhancing procedural efficiency and increasing the number of daily procedures performed.

The reduction in withdrawal time between the L-menthol and the placebo group (MD = 3.24 s) was not significant. This is not concerning, as the U.S. Multi-Society Task Force [83] recommends a withdrawal time of 6–10 min for an effective colonoscopy.

Adverse events were systematically reviewed, with no major, severe, or serious events reported. Commonly reported adverse events included abdominal pain, nausea, diarrhea, and heartburn. Some studies also reported cardiovascular changes, such as electrocardiogram ST-T changes, palpitations, and ventricular premature beats; however, these events were not statistically associated with L-menthol use. The similar proportions of adverse events in the L-menthol and placebo groups in both upper endoscopy and colonoscopy suggest a potentially safe usage profile for L-menthol. Although some studies focused only on serious adverse events, others documented every occurrence. The overall results suggest that L-menthol has a favorable safety profile, as it does not result in more adverse events than placebo. This supports its use as a safe alternative to antispasmodic agents for endoscopic procedures.

Four studies reported adverse drug reactions, with a pooled analysis indicating a significantly lower incidence in the L-menthol group. These findings highlight the equivalent safety profile of L-Menthol compared to placebo.

Previous meta-analyses have examined the antispasmodic effects of L-menthol and peppermint oil during endoscopy. Aziz et al. (2020) [87] focused on colonoscopy and included both oral and topical interventions, while You et al. (2020) [88] evaluated L-menthol for suppressing peristalsis primarily during upper endoscopy, with studies mostly from Japan. In contrast, the present study builds on and extends these works by focusing exclusively on topical application during both colonoscopy and upper endoscopy (EGD), incorporating a larger and more recent evidence base, and providing more detailed subgroup and safety analyses.

Based on current clinical evidence, the topical application (spraying directly onto the mucosa during endoscopic procedures) of L-menthol and peppermint oil seems to be reasonable to achieve an optimal antispasmodic effect. For upper gastrointestinal endoscopy, the established regimen is to spray 20 mL of 0.8% L-menthol solution (160 mg) onto the gastric antrum or body via the endoscope’s working channel before examination. The antiperistaltic effect begins within 30–90 s and is sustained throughout the procedure. Additional doses can be administered if peristalsis recurs during more prolonged examination. For colonoscopy, the regimen is 20 mL of 0.8% L-menthol (160 mg) or 50 mL of peppermint oil solution sprayed or injected onto the colonic mucosa, particularly at the cecum, with additional doses as needed for persistent peristalsis. The antispasmodic effect occurs quickly (within 20–40 s) and lasts at least 15–20 min, covering the withdrawal phase. This approach ensures rapid and sustained suppression of peristalsis, facilitating high-quality mucosal visualization and procedural efficiency, with a favorable safety profile.

### 4.1. Strengths and Limitations

This study is the first to comprehensively synthesize data on L-menthol’s effects across various endoscopic procedures, covering outcomes such as the ADR, peristalsis suppression, ease of examination, and adverse events. By including studies on both colonoscopy and upper endoscopy, the analysis provides a holistic approach to the utility of L-menthol across various endoscopic procedures. Subgroup analyses and advanced statistical methods strengthen the findings.

Most studies were conducted in Asia, particularly Japan, and some in North America. This geographical concentration may limit the generalizability of our findings to other populations, as differences in genetics, healthcare systems, and endoscopic practice may influence the efficacy and safety of L-menthol. Further studies from diverse geographic regions are needed to confirm these findings and ensure their applicability to broader patient populations.

However, high heterogeneity was observed for both the adenoma detection rate (ADR, I^2^ = 51%) and proportion of no peristalsis (PNP, I^2^ = 94%) in colonoscopy, limiting the validity of our results. For the ADR, heterogeneity likely arises from differences in patient age and comorbidity profiles, inconsistent sedation protocols, and variability in endoscopist experience—all known to affect detection rates. For PNP, the much higher heterogeneity reflects broader variations in patient age, sedation practices, and subjective differences in endoscopist assessments of peristalsis (details in Appendix A). Approximately one-third of included studies [30,53,54,61,64,65] had inadequate blinding (e.g., single-blind or open-label designs), which may have introduced performance and detection bias, particularly for subjective outcomes such as ease of examination or peristalsis assessment. While some trials [57,63] employed independent committees and reported moderate-to-good inter-observer agreement, most studies did not validate peristalsis grading scales or assess inter-rater reliability. The scale for the ease of examination outcome has not been formally validated, and subjective assessments by unblinded endoscopists may introduce measurement bias. Furthermore, while procedures were performed by experienced endoscopists in many trials, heterogeneity in operator skill and training may have influenced outcomes. Future studies should prioritize standardized training for outcome assessors and report inter-observer agreement metrics to enhance reproducibility. Adverse events were only assessed during or shortly after the procedure; analysis of any delayed complications or mucosal injury was not possible due to a lack of data. The relatively small sample size of some included studies limits the generalizability and robustness of the conclusions. Furthermore, it was not possible to perform subgroup analyses by age or gender, as the included studies did not provide stratified outcome data [89]. Future studies should assess whether L-menthol’s effects differ across demographic groups, such as older adults or between genders, to clarify potential differences in efficacy or safety. Four studies were sponsored by a pharmaceutical company [57,58,59,63], while others were funded by academic or hospital sources [52,53,56,71,73]. Two studies reported no funding [51,55], the remaining studies did not report their funding sources [30,54,60,61,62,64,65,72]. Details are summarized in Appendix A. The cost effectiveness of L-menthol use was not studied due to a lack of available data; it should be addressed in future research. It is also important to note that despite our broad inclusion criteria, we could not perform meta-analyses for ERCP and EUS due to a lack of available studies. This limitation highlights the need for further research into L-menthol’s efficacy in these endoscopic procedures.

### 4.2. Implications for Practice and Research

Translating scientific findings into clinical practice is essential to improve patient care and achieve better healthcare outcomes [90,91]. Using L-menthol in routine clinical practice may enhance examination quality, increase procedural efficiency, and reduce operator fatigue. Developing guidelines for standardizing L-menthol use in endoscopic protocols will be crucial to maximizing its clinical benefits. Future research should focus on standardizing methodologies, including dosing regimens and outcome measurement. Larger, multicenter trials are needed to confirm the efficacy of L-menthol and determine its optimal clinical application. With these considerations, L-menthol may play a pivotal role in improving the quality and efficiency of gastrointestinal endoscopic procedures.

## 5. Conclusions

L-Menthol may have beneficial effects in gastrointestinal endoscopy, particularly in reducing peristalsis and potentially facilitating easier examinations without an observed increase in adverse events. Although its impact on the ADR remains inconclusive, probably due to variability in study designs and procedural factors, its possible antiperistaltic benefits and improved ease of examination highlight its potential for application in both colonoscopy and upper endoscopy. In addition, L-menthol offers a promising alternative antispasmodic option for patients who cannot receive pharmaceutical agents due to the risks of adverse effects or comorbidities.

## Figures and Tables

**Figure 1 jcm-14-04296-f001:**
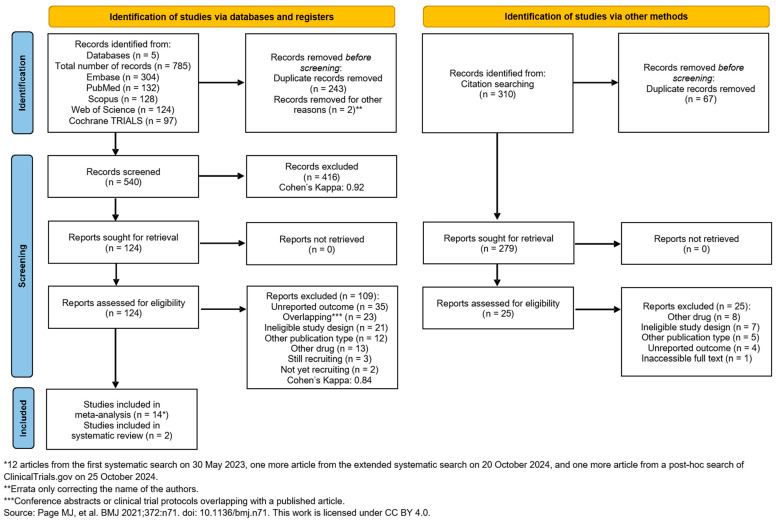
PRISMA 2020 flow diagram for searches in databases and other sources [25].

**Figure 2 jcm-14-04296-f002:**
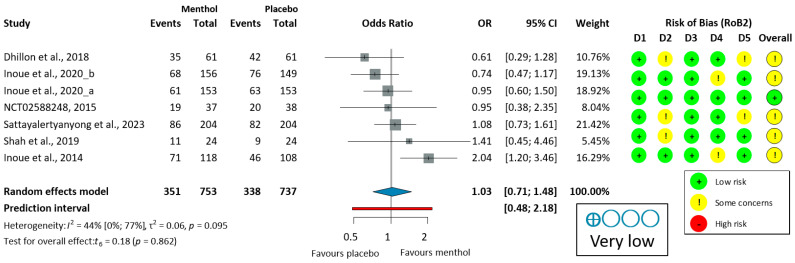
Forest plot of adenoma detection rate in colonoscopy showing pooled ORs with 95% CIs for menthol versus placebo. Risk-of-bias assessments are shown as colored circles for each domain and overall. “Very low” indicates GRADE certainty of evidence. OR = odds ratio, CI = confidence interval, D = domain [51,52,53,54,55,56].

**Figure 3 jcm-14-04296-f003:**
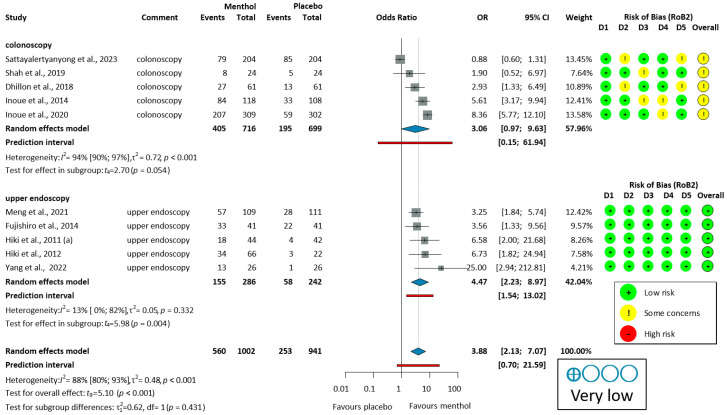
Forest plot of the proportion of no peristalsis (PNP) in colonoscopy and upper endoscopy showing pooled ORs with 95% CIs for menthol versus placebo. Risk-of-bias assessments are shown as colored circles for each domain and overall. “Very low” indicates GRADE certainty of evidence. OR = odds ratio, CI = confidence interval, D = domain [51,53,54,55,56,57,58,59,60,62].

**Figure 4 jcm-14-04296-f004:**
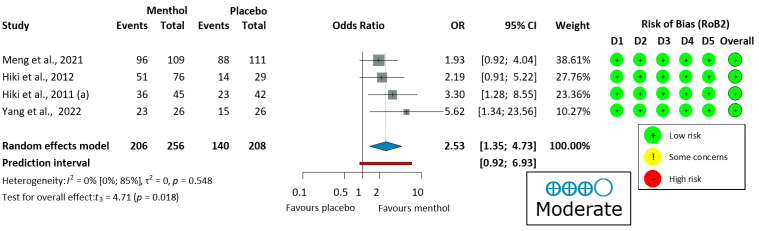
Forest plot of ease of examination for the operator in upper endoscopy showing pooled ORs with 95% CIs for menthol versus placebo. Risk-of-bias assessments are shown as colored circles for each domain and overall. “Moderate” indicates GRADE certainty of evidence. OR = odds ratio, CI = confidence interval, D = domain [58,59,60,62].

**Figure 5 jcm-14-04296-f005:**
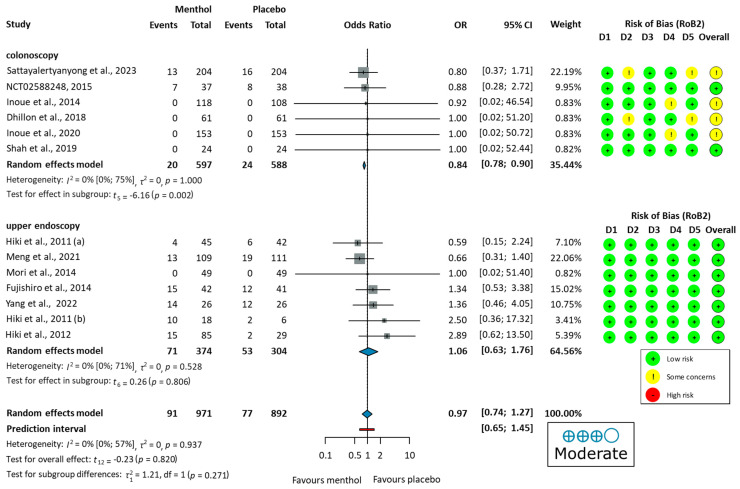
Forest plot of adverse events showing pooled ORs with 95% CIs for menthol versus placebo. Risk-of-bias assessments are shown as colored circles for each domain and overall. “Moderate” indicates GRADE certainty of evidence. OR = odds ratio, CI = confidence interval, D = domain [51,52,53,54,55,56,57,58,59,60,61,62,63].

**Figure 6 jcm-14-04296-f006:**
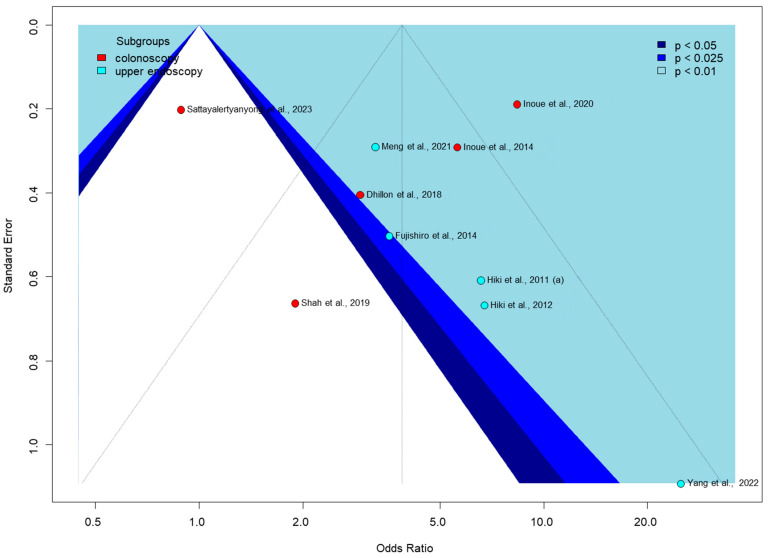
Funnel plot of proportion of no peristalsis [51,53,54,55,56,57,58,59,60,62].

**Figure 7 jcm-14-04296-f007:**
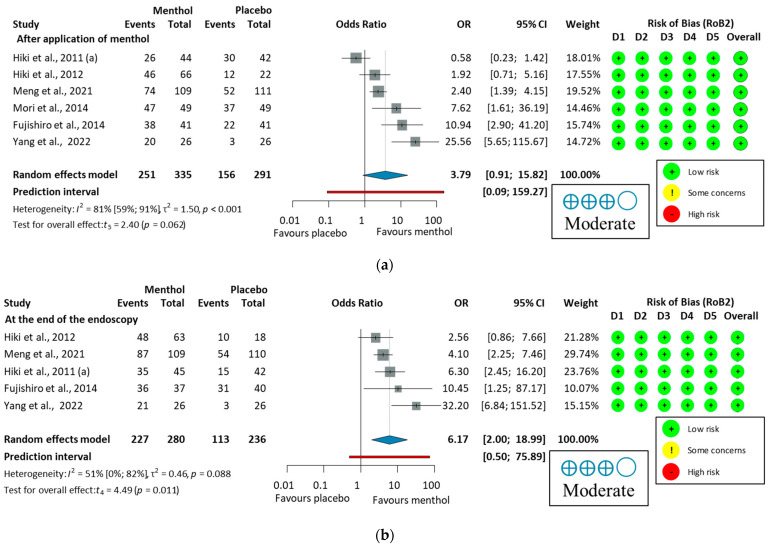
Forest plot of the proportion of peristalsis at different time points in upper endoscopy, (**a**), after application of L-menthol and (**b**), at the end of the endoscopy, showing pooled ORs with 95% CIs for menthol versus placebo. Risk-of-bias assessments are shown as colored circles for each domain and overall. “Moderate” indicates GRADE certainty of evidence. OR = odds ratio, CI = confidence interval, D = domain [57,58,59,60,61,62].

**Table 1 jcm-14-04296-t001:** Baseline characteristics of the RCTs with colonoscopy and upper endoscopy patients investigated in the systematic review and meta-analysis.

Study	(1) Country, (2) Number of Centers,(3) Registration Number	Study Type	Endoscopy Type	Purpose of Endoscopy	Criteria Used to Evaluate Peristalsis	Sample Size (n)	Intervention Type, Dosage	Application Site	Outcomes
Al Moussawi et al., 2017 [72] 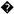	Lebanon,1,NI	double-blind RCT	colonoscopy	diarrhea, bloody diarrhea, iron deficiency anemia, abdominal pain, rectorrhagia	degree of colonic spasm (no movement, minimal, mild, moderate, or marked)	I: 39C: 39	I: Colpermin^®^ 374 mgC: placebo comprised of vitamin B12	orally	PNP, EIE, TAE
Asao et al., 2001 [30]	Japan,1,NI	cohort	colonoscopy	positive test for occult blood infeces, abnormal barium enema, or surveillance afterprevious polypectomies	NI	I: 409C: 36	I: peppermint oil solutionC: placebo	cecum	ADR, TAE,
Dhillon et al., 2018 [51] ‡	Canada,1,NI	double-blindRCT	colonoscopy	screening	NI	I: 61C: 61	I: L-menthol solution;C: placebo (water +simethicone)	cecum	ADR, PNP
Fujishiro et al., 2014 [57]	Japan,8,NCT01411176	double-blindRCT	upper endoscopy	determine the treatment strategy (EMR or ESD)	the modified version of Niwa’sClassification	I: 42C: 41	I: L-menthol solution; 160 mg (0.8%, 20 mL);C: placebo	gastric antrum	PNP, PNMP, TAE, TADR
Han et al., 2021 [73] 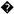	USA,1,NA	double-blindRCT	colonoscopy	screening, positivefecal occult blood test or fecal immunochemical test, surveillance for history of colorectal polyps	classification of colonic peristalsis (0–2)	I: 102C: 90	I: IBGard™ 180 mgC: placebo containing sucrose	orally	ADR, EIE, WT, TAE
Hiki et al., 2011 [58]	Japan,6,NCT00742599	double-blindRCT	upper endoscopy	requiredtreatment or follow-up for confirmed or suspected upperGI disease	the modified versionof Niwa’s Classification	I: 45C: 42	I: L-menthol solution; 160 mg (0.8%, 20 mL);C: placebo	gastric mucosa/antrum	PNP, PNMP, EIE, TAE, TADR
Hiki et al., 2011 [63]	Japan,6,NI	double-blindRCT	upper endoscopy	assessing the tolerability and pharmacokinetics of the intervention	NI	I1: 6I2: 6I3: 6C: 6	I: L-Menthol solution;80 mg (10 mL)160 mg (20 mL)320 mg (40 mL)C: placebo	gastric mucosa	GPPM, TAE
Hiki et al., 2012 [59]	Japan,multi,NI	double-blindRCT	upper endoscopy	required gastric endoscopy	Niwa’s Classification	I: 87C: 29	I: L-menthol solution; 80 mg (0.4%, 20 mL); 160 mg (0.8%, 20 mL); 320 mg (1.6%, 20 mL) C: placebo	gastric antrum	PNP, PNMP, EIE, TAE, TADR
Imagawa et al., 2012 [64] 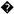	Japan,2,UMIN 000004710	non-randomized prospectivestudy	upper endoscopy	scheduled to undergo EGD were recruited	Niwa’s Classification	I: 1893C: 156	I: L-menthol solution; 1.6%, (20 mL) C: placebo	gastric antrum	AS
Inoue et al., 2014 [53]	Japan,1,UMIN000007972	single-blindprospective RCT	colonoscopy	screening, positive fecal occult blood test follow-up, or postendoscopic resection surveillance	classification of colonic peristalsis (0–3)	I: 118C: 108	I: L-menthol solution; 320 mg (1.6%, 20 mL); C: placebo (water + dimeticone)	cecum	ADR, PNP, PNMP, WT, TAE, TADR
Inoue et al., 2020 [54]	Japan,1,UMIN000023383	single-blindprospective RCT	colonoscopy	screening, positive fecal occult blood test follow-up, or postendoscopic resection surveillance	classification of colonicperistalsis (0–3)	I: 309C: 302	I: L-menthol solution + CO2/air; 160 mg(0.8%, 20 mL)C: placebo (water + dimeticone) + CO2/air	cecum	ADR, PNP, PNMP, WT, TAE
Meng et al., 2021 [60]	China,5,NCT03263910	double-blindRCT	upper endoscopy	advised forUE examination or follow-up for confirmed orsuspected upper GI disease	modified version of Niwa’sclassification	I: 109C: 111	I: L-menthol solution; 160 mg (0.8%, 20 mL); C: placebo	gastric mucosa	PNP, PNMP, EIE, TAE, TADR
Mori et al., 2014 [61]	Japan,1,UMIN000010859	prospectiveopen-label RCT	upper endoscopy	scheduled toscreening or follow-up for upper gastrointestinal disease	modified version of Niwa’sclassification	I: 49C: 49	I: L-menthol solution; 160 mg (0.8%, 20 mL); C: placebo (water + dimeticone)	gastric mucosa	PNMP, TAE
NCT02588248, 2015 [52] §	USA,1,NCT02588248	prospective, double-blindRCT	colonoscopy	primary colorectal cancer screening or survillance	NI	I: 37C: 38	I: peppermint oil solution (0.8% L-menthol, 20 mL) + simethiconeC: simethicone	cecum	ADR, WT, TAE
Sattayaler-tyanyong et al., 2023 [55] ‡	Thailand,1,NCT05559814	double-blindRCT	colonoscopy	screening endoscopy	classification of colonic peristalsis (0–3)	I: 204C: 204	I: peppermint oil solution (0.8% L-menthol, 50 mL) + simethiconeC: simethicone	cecum	ADR, PNP, PNMP, EIE, WT, TAE
Shah et al., 2019 [56]	USA,1,NCT03286764	double-blindRCT	colonoscopy	initial screeningcolonoscopy	classification of colonic peristalsis (0–3)	I: 24C: 24	I: peppermint oil solution (0.8% L-menthol, 50 mL)C: placebo (water + simethicone)	cecum	ADR, PNP, PNMP, WT, TAE
Shavakhi et al., 2012 [71] 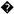	Iran,1,IRCT201107056957N1	prospective, double-blindRCT	colonoscopy	diagnostic or screening colonoscopy	colonic spasm score (0–4)	I: 33C: 32	I: Colpermin^®^ 374 mgC: placebo with lactulose	orally	PNP, TAE
Yang et al.,2022 [62]	Taiwan,1,NCT04593836	prospective, double-blindRCT	upper endoscopy	screening endoscopy	modified version of Niwa’sclassification	I: 26C: 26	I: L-menthol solution; 160 mg (0.8%, 20 mL); C: placebo (olive oil)	gastric mucosa	PNP, PNMP, EIE, TAE
Yoshida et al., 2014 [65] 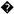	Japan,1,UMIN000008317	retrospective study	colonoscopy	patients with severe colonic spasm	NI	I: 65C: 27	I: 0.8% L-menthol solutionC: water	cecum, ascending colon,transverse colon, descendingcolon, sigmoid colon, rectum,	TAE, PNP

ADR: adenoma detection rate; AS: antispasmodic scores; C: control; EGD: esophagogastroduodenoscopy; EIE: ease of the intragastric examination; EMR: endoscopic mucosal resection; ESD: endoscopic submucosal dissection; GPPM: gastric peristalsis per minute; I: intervention; NI: no information; PNMP: proportion of no or mild peristalsis; PNP: proportion of no peristalsis; TADR: total adverse drug reaction; TAE: total adverse event; WT: withdrawal time. 
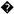
 Study included only in the systematic review. ‡ Conference abstract. § Unpublished study from clinical trial registry with reported results.

## Data Availability

No new data were created or analyzed in this study. Data sharing is not applicable to this article.

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
