# Peer review of "Efficacy and Safety of L-Menthol During Gastrointestinal Endoscopy—A Systematic Review and Meta-Analysis of Randomized Clinical Trials"

_jcm, 2025, doi:10.3390/jcm14124296_

Round 1
Reviewer 1 Report
Comments and Suggestions for Authors
This systematic review and meta-analysis evaluated the safety and efficacy of L-menthol as an antispasmodic agent during gastrointestinal endoscopy, particularly colonoscopy and upper endoscopy. The study found that L-menthol significantly reduced peristalsis and improved procedural ease, but did not increase adenoma detection rate (ADR). While L-menthol appeared safe and well-tolerated, its impact on meaningful clinical outcomes such as lesion detection remains uncertain. The study follows PRISMA 2020 and Cochrane guidelines, demonstrating rigorous methodology.The meta-analysis includes 14 RCTs, offering a relatively robust evidence base for the stated conclusions. The authors thoughtfully address safety, procedural ease, and secondary outcomes beyond ADR.
Although L-menthol reduced peristalsis and improved procedural ease, the absence of improvement in ADR undermines its clinical relevance in colorectal cancer screening. This crucial limitation is under-discussed. A paragraph in the discussion should explicitly state that the primary endpoint (ADR) was not met, and therefore, L-menthol's role as a value-adding intervention in screening colonoscopy is limited. This affects its practical recommendation.
The secondary benefit in upper GI endoscopy, particularly ERCP, where duodenal motility can hinder cannulation, is noted briefly but not elaborated. The authors should explore this more deeply. ERCP procedures may benefit more than routine EGD or colonoscopy from peristalsis suppression, and this could be a future research avenue.
More granular data (e.g., patient age, comorbidities, sedation method) would improve interpretation. Was L-menthol more effective in unsedated patients? Elderly? This could guide clinical targeting of its use.
The authors mention "ease of examination," but it's unclear how this was measured across studies. Was it self-reported? Objective time-based metrics? This should be clarified.
The review reports no significant adverse effects, but it's unclear whether any of the included trials evaluated long-term outcomes or mucosal impact. The authors should mention whether the follow-up period was sufficient to rule out delayed complications.
"promoting relaxation of gastrointestinal smooth muscles via calcium channel blockade" could be reworded to “through calcium channel blockade, L-menthol promotes GI smooth muscle relaxation” for clarity.
“potentially enhancing lesion detection and diagnostic accuracy” → this should be revised to clarify that this potential was not realized, as ADR was unchanged.
Please revise “did not prolong withdrawal time (MD = 3.24 seconds, 95%CI:101.05–107.53)” → these numbers are internally inconsistent; a difference of 3 seconds cannot yield confidence intervals in the 100s. Likely a typo - adjust.
As the authors correctly present, ADR is the single most important quality indicator for colonoscopy and is directly linked to interval cancer risk. Since L-menthol failed to improve ADR, this calls into question its utility in colorectal cancer prevention. This limitation should be acknowledged with greater weight. While comfort and ease of the procedure matter, if detection rates are unchanged, the clinical gain is modest at best.
L-menthol's use in suppressing duodenal motility may be particularly useful in ERCP, where papillary cannulation can be impaired by spasm. The authors briefly mention upper endoscopy but miss the opportunity to explore this. This is a missed chance to emphasize a potential niche indication where the agent could provide clear utility and deserve further study.
So, this is a well-executed review with strong methodology and clear writing. However, the disconnect between procedural benefit and unchanged ADR must be addressed more transparently. Additionally, the opportunity to explore utility in upper GI interventions like ERCP should be expanded. With these clarifications, the paper will provide balanced and actionable guidance to the endoscopy community.
Author Response
Please find our answers in the attached file.

Reviewer 2 Report
Comments and Suggestions for Authors
1) the quality of the trials included in the review was low in many cases. The authors did not remove or test the effect of these low-quality trials. They should have done a sensitivity analysis to see if the results stay the same when only good trials are included. Without this, the results may not be reliable.
2) The authors did not perform a meta-regression to look for causes of the differences between studies. There was a lot of heterogeneity, but they did not explore why. Meta-regression could help explain if study size, procedure type, or population type caused different results. This is a major statistical issue and must be addressed. If they cannot do it now, they should say in the discussion that this is a limitation.
3) Some outcomes like "ease of examination" are very subjective. Different endoscopists may give different scores. There is no explanation if these outcomes were standardized or validated. This brings risk of measurement bias. The authors must say this in the discussion clearly.
4) They also did not talk about how many trials used blinding. Many studies did not blind the participants or the doctors. This can create performance bias, especially for subjective outcomes. It is important that they add this as a limitation.
5) The authors pooled results from different procedures like colonoscopy and gastroscopy, but these procedures are very different. The effect of L-menthol may not be the same in each. They should have separated the analysis or at least discussed this problem. Now it creates a clinical confounder. Also, they did not include any subgroup analysis by age or gender. For example, older patients may react differently to L-menthol. This must be added as a limitation.
6) The confidence intervals in many results were very wide and even crossed the null. This means the results are not statistically significant. Still, the authors used strong language to say L-menthol was helpful. This is misleading. They must change the text to say the evidence is weak or unclear. The GRADE tables also show that the certainty of evidence was mostly “very low” or “moderate.” The authors often ignore this and write as if the results are strong. The authors must write honestly and clearly in the discussion.
7)They also did not check if the trials were sponsored by companies. If a drug company funded the study, the results can be biased. The authors must check and report the funding source of each trial. If they do not know, they must say that and list it as a limitation.
8) Also, they did not analyze the cost or cost-effectiveness of using L-menthol. This is important for doctors and hospitals. If they cannot calculate this now, they should say that cost was not studied and must be explored in the future.
9) There is also a missing part about inter-observer agreement. When many doctors rate outcomes like peristalsis, they must agree on what they see. But this review does not mention if the trials had any calibration or training. This makes the results hard to trust. It must be added as a limitation too.
10) Another point is that the search strategy was very wide, but the authors did not explain if they found studies in all regions or just some countries. This can cause publication bias or geographical bias. If the trials are mostly from Asia or Europe, the results may not apply to all populations. They must say where the trials came from and add that this may limit generalization.
11) the authors must stop using strong or confident words in the conclusion. This paper shows a possible effect of L-menthol, but with weak evidence. The article should say that more high-quality studies are needed and that the current evidence is not strong. All the statistical problems, bias, and missing subgroup analyses must be written in the discussion section as limitations.
Author Response
Please see our answers in the attached file.

Round 2
Reviewer 2 Report
Comments and Suggestions for Authors
The authors answered all the points in a clear and technical way. Their answers were complete and followed what was asked in the review.